# Coupling nitrate electrochemical reduction and nitrite oxidation of ethanol for acetamide synthesis

Qikun Hu[1,4], Ouwen Peng [ID][1,4], Jia Liu[1,4], Mengyao Su[1], Junyuan Feng[2], Kun Zhang[1], Derong Chen [ID][1], Zong-Xiang Xu [ID][2] & Kian Ping Loh [ID][1,3] ✉

Electrochemical acetamide synthesis under ambient conditions offers a sustainable route for converting waste nitrate into valuable chemicals. Conventional methods, limited to standalone reduction or oxidation processes, typically achieve low Faradaic efficiencies (<40%) and yields (<0.2 mmol h$^{-1}$ cm$^{-2}$). Here, we present a tandem reaction system coupling cathodic reduction and anodic oxidation in a full-cell electrolyzer to enhance acetamide production. At the cathode, nitrate is first reduced to nitrite, which subsequently oxidizes ethanol to acetaldehyde. This acetaldehyde reacts in situ with electrogenerated ammonia to form α-aminoethanol. The intermediate is then transported to the anode, where it undergoes oxidation to yield acetamide. The reaction pathway is confirmed through proton nuclear magnetic resonance spectroscopy, revealing efficient acetamide synthesis with a yield of 7.2 ± 0.3 mmol h$^{-1}$ (0.45 ± 0.02 mmol h$^{-1}$ cm$^{-2}$) at a cell voltage of 2.4 V. Furthermore, the strategy extends to other amides, such as formamide and butyramide, underscoring its versatility. Techno-economic analysis highlights the viability of this route, with estimated production costs competitive against conventional thermal processes.

Amides are highly valuable as versatile feedstocks for producing industrial solvents and pharmaceutical intermediates[1–6]. Currently, ambient electrosynthesis of amides from waste-derived nitrogen sources (e.g., nitrate and nitrite) proceeds through two conventional pathways[6,7]. The first strategy couples the nitrate reduction reaction (NO$_3$RR) with the CO$_2$/CO reduction reaction (CO$_2$RR/CORR) at the cathode[8,9]. However, the markedly different onset potentials and kinetics of NO$_3$RR and CO$_2$RR/CORR[10–13] make it difficult to synchronize their rates, resulting in low Faradaic efficiency and poor acetamide yields. The second strategy involves the ethanol oxidation reaction (EOR) in concentrated ammonia solutions at the anode[14,15], typically paired with hydrogen evolution at the counter electrode. In this case, severe competition between ammonia oxidation and ethanol oxidation[16] leads to significant ammonia loss and similarly low acetamide productivity.

Because amide formation requires complementary carbon and nitrogen sources to form the C-N bond[17], current acetamide electrosynthesis methods—which rely on separate oxidation and reduction steps—are limited by competing side reactions and inherently low efficiencies.

In this work, we designed a tandem pathway that captures an intermediate generated during cathodic nitrate reduction (e.g., nitrite) and uses it to chemically oxidize ethanol, thereby generating reactive intermediates that feed back into the electrochemical process. By using a chemical step to generate the intermediate, the system decouples the carbon activation from the electrode potential. This removes the kinetic competition and allows the cathode to operate at its most efficient potential for nitrate reduction. Figure 1 presents the overall scheme: C-N coupling at the cathode forms α-aminoethanol,

[1]Department of Chemistry, National University of Singapore, 3 Science Drive 3, Singapore, Singapore. [2]Department of Chemistry, Southern University of Science and Technology, Shenzhen, China. [3]Centre for Hydrogen Innovations, National University of Singapore, E8, 1 Engineering Drive 3, Singapore, Singapore. [4]These authors contributed equally: Qikun Hu, Ouwen Peng, Jia Liu. ✉e-mail: chmlohkp@nus.edu.sg

which is subsequently oxidized to acetamide at the anode. Cathodic nitrate reduction to nitrite is coupled to the ethanol to acetaldehyde oxidation sequence. Ammonia produced from nitrate reduction then reacts with acetaldehyde to yield α-aminoethanol. We found that initiating the process from the anode (Fig. S1) promotes acetate formation over acetaldehyde, significantly reducing the acetamide yield.

To enhance α-aminoethanol adsorption and promote oxidation to acetamide, we synthesized a Rh single-atom–modified Ni(OH)$_2$ catalyst (Rh$_1$/Ni(OH)$_2$) for the anode. With this design, the system achieves a high acetamide production rate of $7.2 \pm 0.3$ mmol h$^{-1}$ ($0.45 \pm 0.02$ mmol h$^{-1}$ cm$^{-2}$) at a cell voltage of 2.4 V, and the approach is extendable to formamide and butyramide synthesis.

The formation of α-aminoethanol at the cathode was confirmed using proton nuclear magnetic resonance ($^1$H NMR). In addition, a techno-economic analysis (TEA) indicated that this method is economically viable, supported by the high commercial value of acetamide.

Overall, this work demonstrates an efficient integrated strategy for acetamide electrosynthesis that merges chemical and electrochemical steps, offering a promising pathway toward full-cell tandem amide production.

## Results
### Characterizations and catalytic performance of the Rh$_1$/Ni(OH)$_2$ catalyst

The key to efficient full-cell operation is to use catalysts with high performance for NO$_3$RR and EOR. We selected ruthenium nanoparticles-decorated Cu$_2$O nanowires (Ru/Cu$_2$O) catalyst for NO$_3$RR, and synthesized it according to reported procedures[12,18]. Ni(OH)$_2$ on Ni foam, synthesized by hydrothermal synthesis, was used as the anode[19]. The Ni(OH)$_2$ sheets exhibited crystalline facets under scanning electron microscope (SEM, Fig. S2). Rh$_1$/Ni(OH)$_2$ was fabricated by Ni(OH)$_2$ impregnation with Rh(CH$_3$COO)$_3$ ethanol solution and subsequent annealing in a hydrogen/argon mixture. The metal loading of Rh single atoms on Ni(OH)$_2$ was determined as 0.4 wt% from inductively coupled plasma optical emission spectroscopy (ICP-OES).

A Ni(OH)$_2$ (100) facet with Rh single atoms can be observed in high-angle annular dark-field scanning transmission electron microscopy (STEM-HAADF, Fig. 2a, b). Fast Fourier transform (FFT) of Fig. 2a of the diffraction spots confirms that the main diffraction spots of Ni(OH)$_2$ originate from (100) facets (Fig. S3). The valence state and coordination environment of Rh$_1$/Ni(OH)$_2$ were examined by X-ray absorption spectroscopy (XAS) and X-ray photoelectron spectroscopy (XPS). Rh $K$-edge X-ray absorption near-edge structure (XANES) spectrum of Rh$_1$/Ni(OH)$_2$ was collected to reveal its oxidation state (Fig. S4). Extended X-ray

absorption fine structure (EXAFS) spectroscopy confirms the existence of Rh single atoms on the Rh$_1$/Ni(OH)$_2$ catalyst surface, with a prominent Rh-O peak at 1.5 Å was observed in the Fourier transform-EXAFS spectra in Fig. 2c[20,21]. Similar to the results in XANES, X-ray photoelectron spectroscopy of Rh single atoms in Rh$_1$/Ni(OH)$_2$ catalyst reveals 309.2 and 314.2 eV spin-orbit doublets in the Rh$_{3d}$ spectrum (Fig. S5), which can be attributed to positively charged Rh species (Rh$^{3+}$)[22,23]. We have also examined the electrochemically active surface area (ECSA) of Ni(OH)$_2$ and Rh$_1$/Ni(OH)$_2$ catalysts (Fig. S6). The presence of Rh single atom slightly increases the active area of Ni(OH)$_2$.

The calibration curves of all the products tested are shown in Figs. S7, S8. First, we evaluated the ethanol oxidation performance of Rh single atom modified Ni(OH)$_2$. As shown in Fig. S9, Rh$_1$/Ni(OH)$_2$ catalyst achieves a much higher current density and acetate Faradaic efficiency than the bulk Ni(OH)$_2$ catalyst. Considering that both ammonia and acetate are raw products for acetamide synthesis (i.e., ammonium acetate can be dehydrated to form acetamide)[24], we coupled NO$_3$RR in parallel with EOR in a 16 cm$^2$ flow electrolyzer. Ru/Cu$_2$O and Rh$_1$/Ni(OH)$_2$ catalysts were employed at the cathode and anode, respectively. About 1 M nitrate and 1 M ethanol water solution at pH 14 were used as catholyte and anolyte with a 3 mL min$^{-1}$ flow rate, respectively. As shown in Fig. 2d, this system achieves 10.0 A current at 2.4 V cell voltage, synthesizing ammonia at 90.7% Faradaic efficiency with 39.7 mmol h$^{-1}$ yield rate and acetate at 81.6% Faradaic efficiency with 71.6 mmol h$^{-1}$ yield rate (Fig. 2e, f), which is significantly higher than using Ni(OH)$_2$ as anode (Fig. S10). The total concentration of N-containing species was kept constant during the whole reaction (Fig. S11). These benchmark performances demonstrate the potential of our system to provide feedstocks for industrial acetamide synthesis. Lower-Rh loading of 0.1 wt% and higher Rh loading of 1 wt% (Rh nanoparticles, Fig. S12) on Ni(OH)$_2$ were also synthesized, showing poorer performance in ethanol oxidation (Fig. S13).

DFT simulations were performed to provide insights into the role of Rh single atoms on Ni(OH)$_2$. The charge density difference of Rh$_1$/Ni(OH)$_2$ catalyst is displayed in Fig. S14a. Through bonding with the O atoms on the Ni(OH)$_2$ surface, Rh single atoms become positively polarized, which promotes its ability to bind with the hydroxyl group in alcohol. This is also confirmed in projected crystal orbital Hamilton population (pCOHP) analysis (Fig. S15a). Compared to the Ni atom in bulk Ni(OH)$_2$, the Rh single atom exhibits a higher bonding integral with ethanol arising from its stronger Rh $4d_{xz}$ · O $2p_z$ interactions. Therefore, Rh$_1$/Ni(OH)$_2$ catalyst has stronger ethanol adsorption (−1.04 vs. −0.38 eV) compared to Ni(OH)$_2$ catalyst (Fig. S14b). Rh single atoms also facilitate subsequent dehydrogenation of adsorbed ethanol

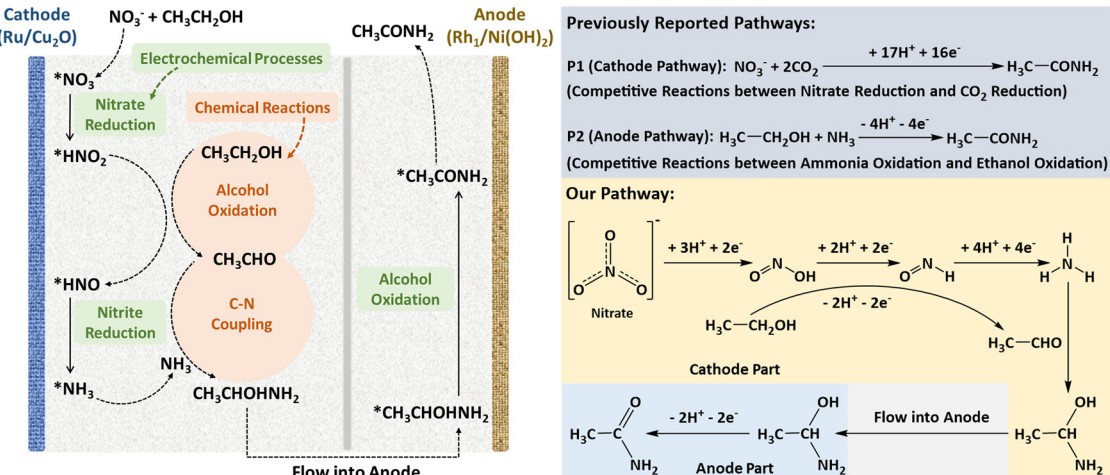

**Fig. 1 | Schematic illustration.** Full cell reaction pathway for direct acetamide electrosynthesis based on tandem reaction at the cathode (electrochemical + chemical) and anode (electrochemical). The left schematic shows the difference between reported pathways utilizing only anode or cathode and our full-cell reaction pathway.

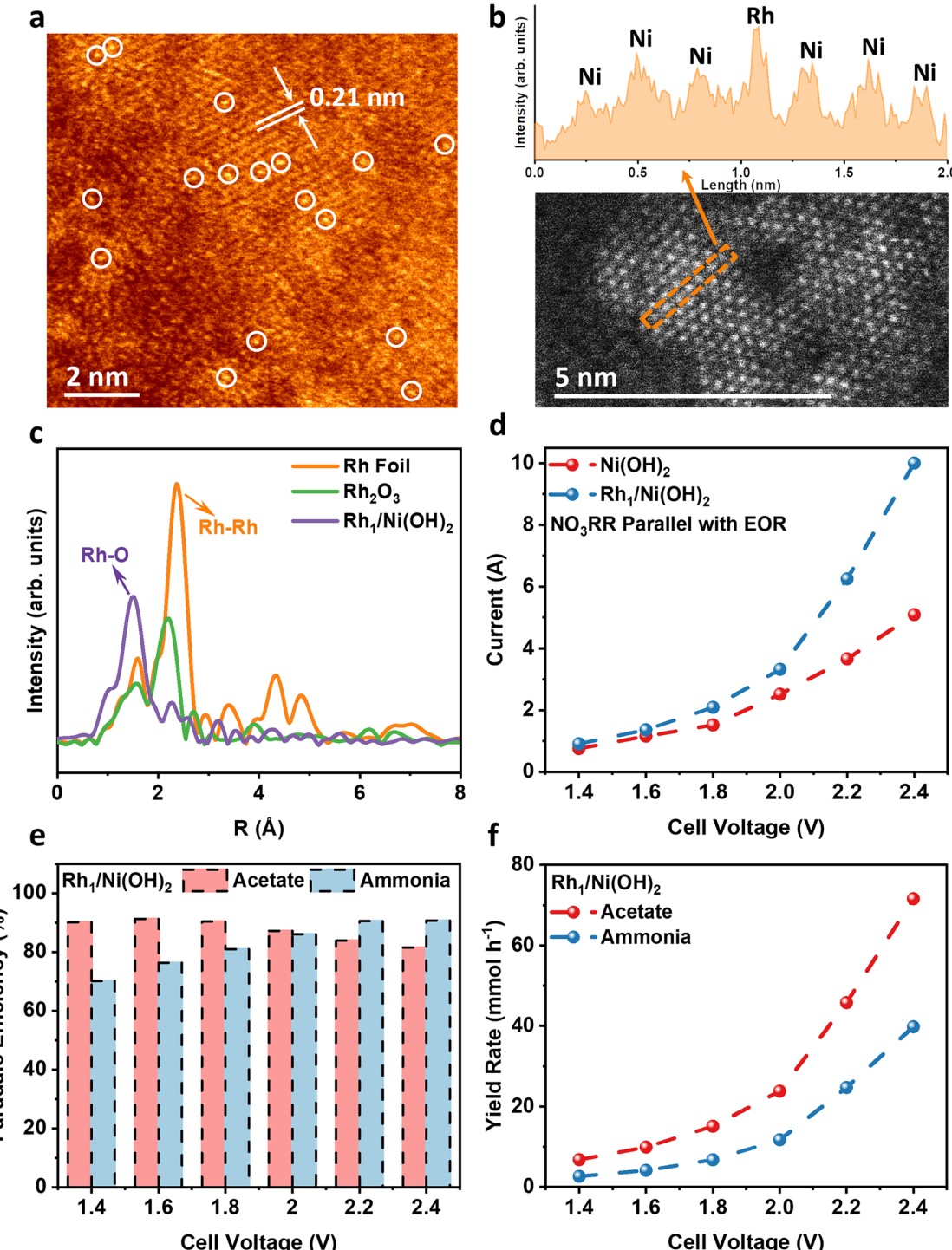

**Fig. 2 | Characterizations and catalytic performance of the Rh₁/Ni(OH)₂ catalyst. a** STEM-HAADF image of Rh₁/Ni(OH)₂; **b** Intensity profiles of one Rh single atom; **c** Rh *K*-edge EXAFS spectra of various catalysts; **d** Current of different anode catalysts in the 16 cm² flow electrolyzer coupling NO₃RR and EOR in parallel; **e** Faradaic efficiency and **f** products yield rate of products in the 16 cm² flow electrolyzer applying Rh₁/Ni(OH)₂ catalyst as anode and Ru/Cu₂O as cathode. Scale bar: **a** 2 nm; **b** 5 nm.

through weakening of the O-H bond (Fig. S15b), reducing the energy barriers for ethanol oxidation to acetate.

### Direct acetamide electrosynthesis by coupling nitrate electrochemical reduction with oxidation of ethanol by in situ-generated nitrite

We designed a reaction pathway to directly synthesize acetamide by coupling both chemical reactions and electrochemical processes. Ru/Cu₂O and Rh₁/Ni(OH)₂ catalysts were chosen as cathode and anode,

respectively. As depicted in Fig. 3a, unlike previous experimental set-ups (NO₃RR and EOR work in parallel), we prepared an electrolyte containing both nitrate and ethanol and flowed it from the cathode to the anode to couple nitrate reduction and ethanol oxidation in series. We proposed that, during nitrate reduction, a significant amount of nitrite is produced as a key intermediate[25], which then oxidizes ethanol to acetaldehyde[26]. After that, the ammonia generated from nitrate reduction can react with acetaldehyde to achieve C-N coupling for the formation of α-aminoethanol at the cathode[27]. The α-aminoethanol

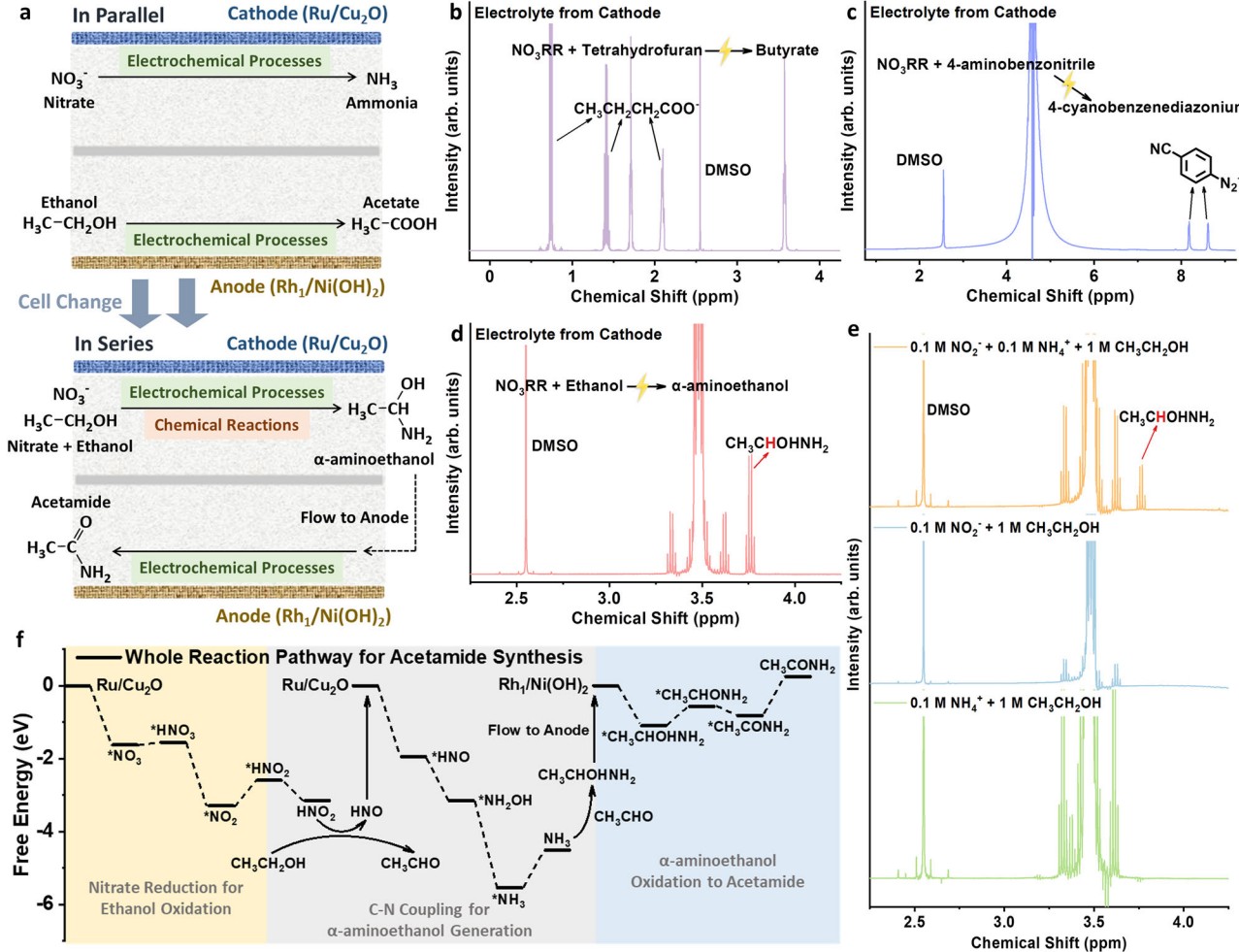

**Fig. 3 | Direct acetamide electrosynthesis by coupling nitrate electrochemical reduction with oxidation of ethanol by in situ-generated nitrite. a** Schematic illustration of parallel and series (Tandem) reaction pathways; **b** $^1$H NMR validation of butyrate generation at the cathode under 2.4 V cell voltage; **c** $^1$H NMR validation of 4-cyanobenzenediazonium generation at the cathode under 2.4 V cell voltage; **d** $^1$H NMR validation of α-aminoethanol generation at cathode under 2.4 V cell voltage; **e** $^1$H NMR validation of α-aminoethanol produced by mixing different reactants in water; **f** Whole reaction pathway for acetamide electrosynthesis by coupling nitrate reduction and ethanol oxidation in tandem.

then flows to the anode, where it is oxidized to acetamide by Rh$_1$/Ni(OH)$_2$ catalyst. Differential electrochemical mass spectrometry (DEMS) revealed that, at the same current density (25 mA cm$^{-2}$), ethanol addition to the NO$_3$RR system resulted in a smaller NO signal on the catalyst surface (m/z = 30, an intermediate from HNO$_2$ to HNO), supporting our proposed reaction mechanism (Fig. S16).

The strong oxidation ability of in situ-generated nitrite from nitrate electrochemical reduction were examined by adding reactants such as tetrahydrofuran (THF) and 4-aminobenzonitrile oxidation during NO$_3$RR, where higher value products such as butyrate and 4-cyanobenzenediazonium were obtained in the cathode electrolyte, respectively (Fig. 3b, c and Figs. S17–S20)[28–30]. To enhance product value, we added ethanol in NO$_3$RR to facilitate C-N coupling at the cathode. The generation of the C-N coupling product from the cathode, α-aminoethanol, can be verified through $^1$H NMR detection of the cathode electrolyte (Fig. 3d). A further proof of this pathway is that simply mixing nitrite, ammonium, and ethanol in 0.1 M KOH water solution produces α-aminoethanol, as judged from its signal in $^1$H NMR (Fig. 3e). However, if nitrite or ammonium is absent, no α-aminoethanol signals can be observed in $^1$H NMR. This highlights that the co-existence of nitrite and ammonium with ethanol are necessary for α-aminoethanol formation (Figs. S21–S23). We have also conducted DFT simulations to validate the reliability of the whole reaction pathway (Fig. 3f). No significant energy barriers were observed.

## Efficient acetamide electrosynthesis in the 16 cm$^2$ flow electrolyzer

We evaluated the acetamide synthesis performance of our pathway in a 16 cm$^2$ flow electrolyzer. The electrolyte pH was 13 (0.1 M KOH), and the electrolyte after reaction was bubbled with CO$_2$ to avoid product decomposition (Fig. S24). α-aminoethanol intermediate was detected in the cathode electrolyte during the electrocatalytic process (Fig. S25). Subsequently, the electrosynthesized acetamide was identified and quantified from the anode electrolyte by $^1$H NMR (Figs. S26, S27). Various electrolyte flow rates were examined in our system. While higher flow rates enhance the Faradaic efficiency for nitrite (Fig. S28a), they also dilute nitrite and ammonia concentrations, resulting in decreased acetamide yield (Fig. S28b). Therefore, 1.5 mL min$^{-1}$ was identified as the optimal flow rate to fully utilize electrogenerated nitrite and ammonia. Under this flow rate, the Faradaic efficiency of products (nitrite and ammonia) from NO$_3$RR without ethanol addition is shown in Fig. S28c.

In the tandem reaction pathway, the α-aminoethanol synthesized at the cathode flows to the anode to be oxidized to acetamide at the anode. As illustrated in Fig. 4a, a high yield of acetamide (7.2 ± 0.3 mmol h$^{-1}$, 0.45 ± 0.02 mmol h$^{-1}$ cm$^{-2}$) was achieved at a cell voltage of 2.4 V, with 89 ± 1% Faradaic efficiency (Fig. 4b). Only ~10 kWh kg$^{-1}$ electricity power was required for acetamide synthesis (Fig. 4c). Our system can also be used for formamide and butyramide synthesis, with up to 3.5 ± 0.2 and 3.4 ± 0.4 mmol h$^{-1}$ yield rate,

respectively (Figs. S29–S33). In such a system, direct nitrate reduction to ammonia and alcohol oxidation to carboxylate are the two main side reactions. The side products of ammonia and carboxylate are quantified in Fig. S34. Control experiments using bulk $Ni(OH)_2$ as anode (Fig. S35a) shows that it has poorer acetamide synthesis performance than $Rh_1/Ni(OH)_2$. Using the same electrolyte but reversing the flow direction in a single pass (from anode and ends at cathode) decreased the acetamide yield significantly (Fig. S35b), highlighting the crucial role of ethanol oxidation by nitrite in acetamide synthesis.

The reaction kinetics of our system were analyzed next. As depicted in Fig. S36, the apparent reaction order for ethanol concentration is higher than that for nitrate concentration (1.54 vs. 1.42), suggesting that ethanol concentration exerts a stronger influence on the acetamide yield rate compared to nitrate concentration. This is also reflected in Fig. 4d, whereby increasing the ethanol concentration to 1–1.5 M significantly enhances the acetamide yield rate. However, further increasing the ethanol concentration to 2 M reduces the electrolyte's conductivity, leading to a considerable decrease in the acetamide yield rate. Similarly, our system achieves a high acetamide yield rate with a 1–1.5 M nitrate concentration. However, at 2 M nitrate concentration, the ammonia yield rate decreases slightly, which slightly reduces the acetamide yield rate[12]. Additionally, increasing the flow rate will lead to a −2.01 apparent reaction order, which corresponds to a significant decrease in the acetamide yield rate.

Process durability is another key criterion for the practical amide electrosynthesis. Continuous acetamide synthesis can be carried out in our system at 2.4 V cell voltage for 3000 min. The yield rate of acetamide is stable at -7.1 mmol $h^{-1}$ (Fig. 4e). We vacuum-drained the electrolyte at room temperature after the reaction, then extracted acetamide from the resulting solid using ethyl acetate. The extract was subsequently vacuum-dried again at room temperature to generate purified acetamide. Gas chromatography-mass spectrometry (GC-MS) confirmed the structural integrity of the synthesized acetamide (Fig. S37). Based on the $^1H$ NMR spectra (Fig. S38) and elemental analysis data (Table S1), the synthesized acetamide is verified to be highly pure. These results highlight the potential of our system for practical amide production.

To evaluate the stability of the $Rh_1/Ni(OH)_2$ catalyst, we performed in situ XAS (Fig. S39). During ethanol oxidation, the Ni $K$-edge XANES spectra remained unchanged at current densities of 50 and 100 mA $cm^{-2}$ for 20 and 40 min, indicating that the $Ni(OH)_2$ structure was preserved. Post-stability test analyses using XRD, XPS and XAS further confirmed that there is no change in the coordination environment for the active sites in the catalyst. The XRD pattern showed no structural changes in $Ni(OH)_2$ (Fig. S40a), and Ni $2p$ XPS spectra indicated that the Ni valence state (Fig. S40b) did not change. Rh $K$-edge EXAFS spectra demonstrated that Rh's coordination environment remained stable (Fig. S40c). Additionally, SEM images showed no noticeable morphological changes at the end of the reaction (Fig. S41).

## Technical and economic analyses of acetamide electrosynthesis

Economic feasibility is a crucial aspect when evaluating a reaction pathway[31]. To assess the economic viability of producing acetamide of 95% purity using potassium nitrate and ethanol in a United States (US) factory, we conducted a comprehensive analysis. The simulation, performed in Aspen Plus, encompassed various factors such as raw material treatment, acetamide electrosynthesis, purification and storage, labor costs, instrument maintenance, taxes, marketing expenses, and other relevant parameters[18]. Detailed models and parameters utilized in the simulation are provided in the supporting information. The flowsheet of acetamide electrosynthesis is shown in Fig. S42.

Initially, we examined the influence of certain reaction parameters on the cost of amide production. Figure 5a illustrates the significant impact of the Faradaic efficiency of α-aminoethanol, an important intermediate in amide generation, on production costs. With a cell voltage of 2.4 V, only 55 mA $cm^{-2}$ current density is required to achieve

acetamide market price when the Faradaic efficiency of α-aminoethanol is 30%. When the Faradaic efficiency of α-aminoethanol reaches 52.5%, the financial profitability of acetamide production can be achieved within a broad current density range of 40–160 mA $cm^{-2}$ in a cell voltage range of 1.4–2.4 V. (Fig. 5b). These results underscore the high value of acetamide.

Furthermore, we simulated the actual operation of the factory by fitting experimental data. Figure 5c highlights that during the establishment stage, distillation and extraction equipment constitute the main capital expenditures (CAPEX). In the production phase of acetamide, the costs primarily arise from the terms of entrainer and heating, accounting for the majority of operating expenditures (OPEX). Considering acetamide's value as a product, a factory operating at a cell voltage of 2.4 V would only require three years to recoup the investment and generate profits (Fig. 5d). For formamide and butyramide, it will take nineteen and six years of production, respectively, to recover the costs and start making profits. This finding is further corroborated by the global sensitivity analysis of techno and economic analyses (TEA), which evaluated the impact of the net present value (NPV) variance from different parameters (Fig. 5e). These results provide substantial evidence of the economic feasibility of our pathway for amide electrosynthesis.

## Discussion

In summary, we have demonstrated the electrochemical synthesis of acetamide by coupling chemical reactions and electrochemical processes in a 16 $cm^2$ flow electrolyzer. The coupling mechanism involves reactive intermediates that are generated during the electrochemical process itself. These intermediates (e.g., Nitrite) can act as oxidizing agents for other reactants in the system, effectively creating a dual pathway where both reduction and oxidation occur simultaneously or sequentially within the same system. Specifically, we have demonstrated tandem $NO_3RR$ and EOR, where the in situ-generated nitrite oxidizes ethanol to acetaldehyde, then C-N coupling between acetaldehyde and electrosynthesized ammonia produces α-aminoethanol at the cathode. In tandem, α-aminoethanol is oxidized to acetamide at the anode. Based on this, we achieved an acetamide yield rate of 7.2 mmol $h^{-1}$ (Table S2). Our method of acetamide synthesis represents a significant advancement in sustainable chemical manufacturing. The integration of ethanol oxidation in nitrate reduction exemplifies how chemical reactions can complement electrochemical processes, enhancing overall efficiency and broadening the scope of products.

## Methods

### Synthesis of $Ni(OH)_2$ nanosheets on Ni foam

All chemicals were obtained from Sigma-Aldrich and used as received unless otherwise noted. Ni foam (2-mm thickness, 100 PPI, Latech Scientific Supply Pte. Ltd.) was cleaned by sequential rinsing with 1 M HCl and deionized water. A $4 \times 4$ $cm^2$ piece of the Ni foam was then placed in a Teflon-lined stainless-steel autoclave containing 200 mL of an aqueous solution with 25 mmol $NiCl_2$ and 50 mmol hexamethylenetetramine at room temperature. The autoclave was heated to 100 °C for 10 h and allowed to cool naturally to room temperature. The resulting Ni foam decorated with $Ni(OH)_2$ nanosheets was subsequently washed with water and ethanol.

### Rh single atoms decorated $Ni(OH)_2$ catalyst on Ni foam ($Rh_1/Ni(OH)_2$)

The dried $Ni(OH)_2$-coated Ni foam was immersed in 100 mL of a 0.2 mM $Rh(CH_3COO)_3$ ethanol solution for 2 h (or 30 min for the lower-Rh-loading sample) under stirring at 500 rpm, then rinsed three times with deionized water and dried at room temperature. The sample was subsequently annealed in a hydrogen/argon gas mixture (80:20 sccm) at 200 °C for 2 h with a heating rate of 10 °C $min^{-1}$ to obtain the $Rh_1/Ni(OH)_2$ catalyst.

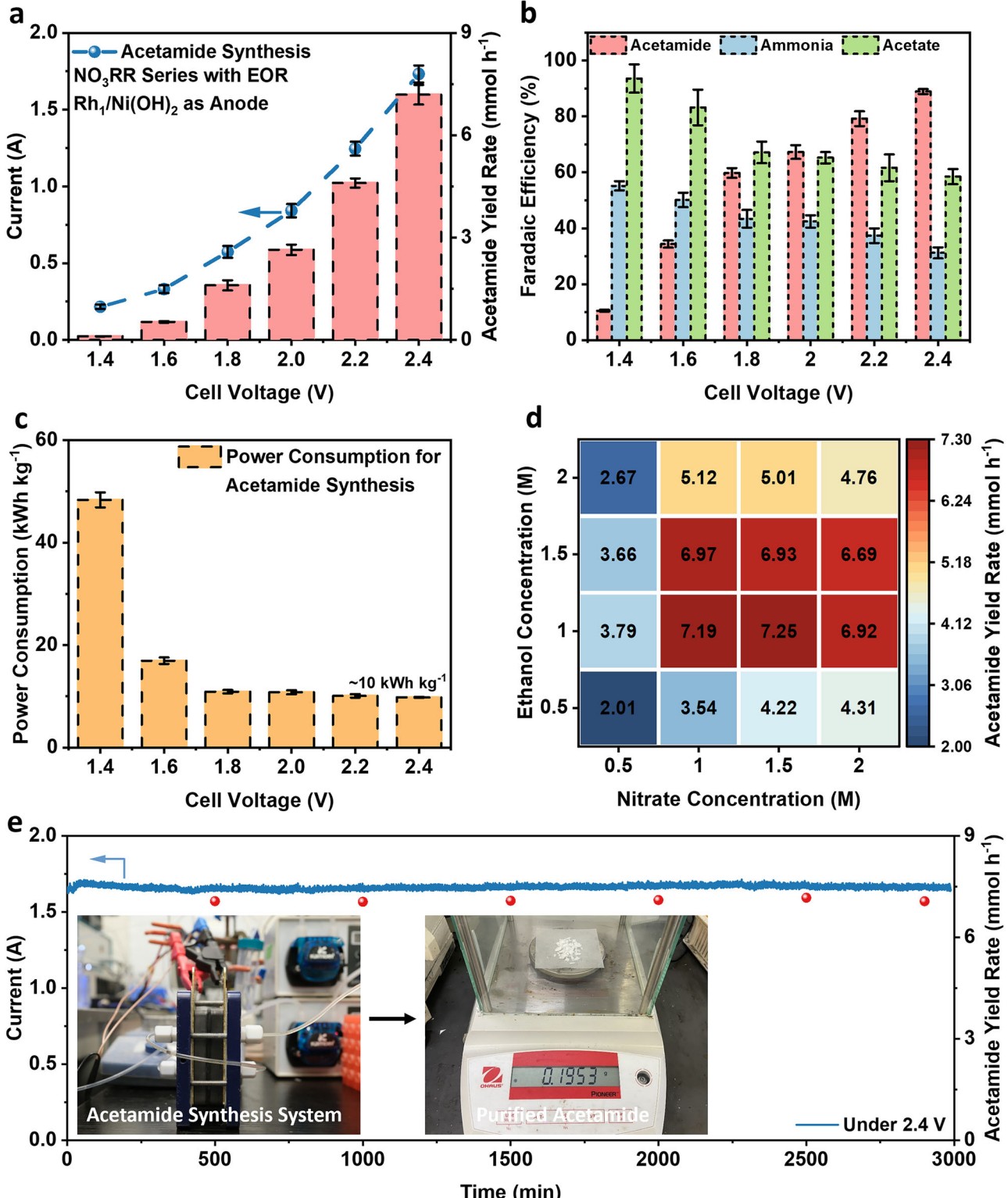

**Fig. 4 | Efficient acetamide electrosynthesis in the 16 cm² flow electrolyzer.**
**a** Current (left axis) and yield rate (right axis) for acetamide electrosynthesis;
**b** Faradaic efficiency (FE) of different products during acetamide electrosynthesis
(based on the sum of FE of both anode and cathode); **c** Power consumption for
acetamide electrosynthesis; **d** Acetamide yield rate dependence on nitrate and
ethanol concentration under 2.4 V cell voltage; **e** Stability test for continuous
acetamide electrosynthesis under 2.4 V cell voltage. Insert: the 16 cm² flow elec-
trolyzer with two peristaltic pumps and purified acetamide (0.1953 g from 45 mL
electrolyte) collected after stability test. Error bars (in standard deviation) are
present for three repetitive experiments.

## Rh nanoparticles-decorated $Ni(OH)_2$ catalyst on Ni foam (Rh Nanoparticles/$Ni(OH)_2$)

The dried $Ni(OH)_2$-coated Ni foam was immersed in 100 mL of a 0.2 mM $Rh(CH_3COO)_3$ aqueous solution for 2 h with stirring at 500 rpm. Afterward, the foam was rinsed three times with deionized water and dried at room temperature. The sample was then annealed in a hydro-gen/argon gas mixture (80:20 sccm) at 200 °C for 2 h, with a heating rate of 10 °C min$^{-1}$, to yield the Rh Nanoparticles/$Ni(OH)_2$ catalyst.

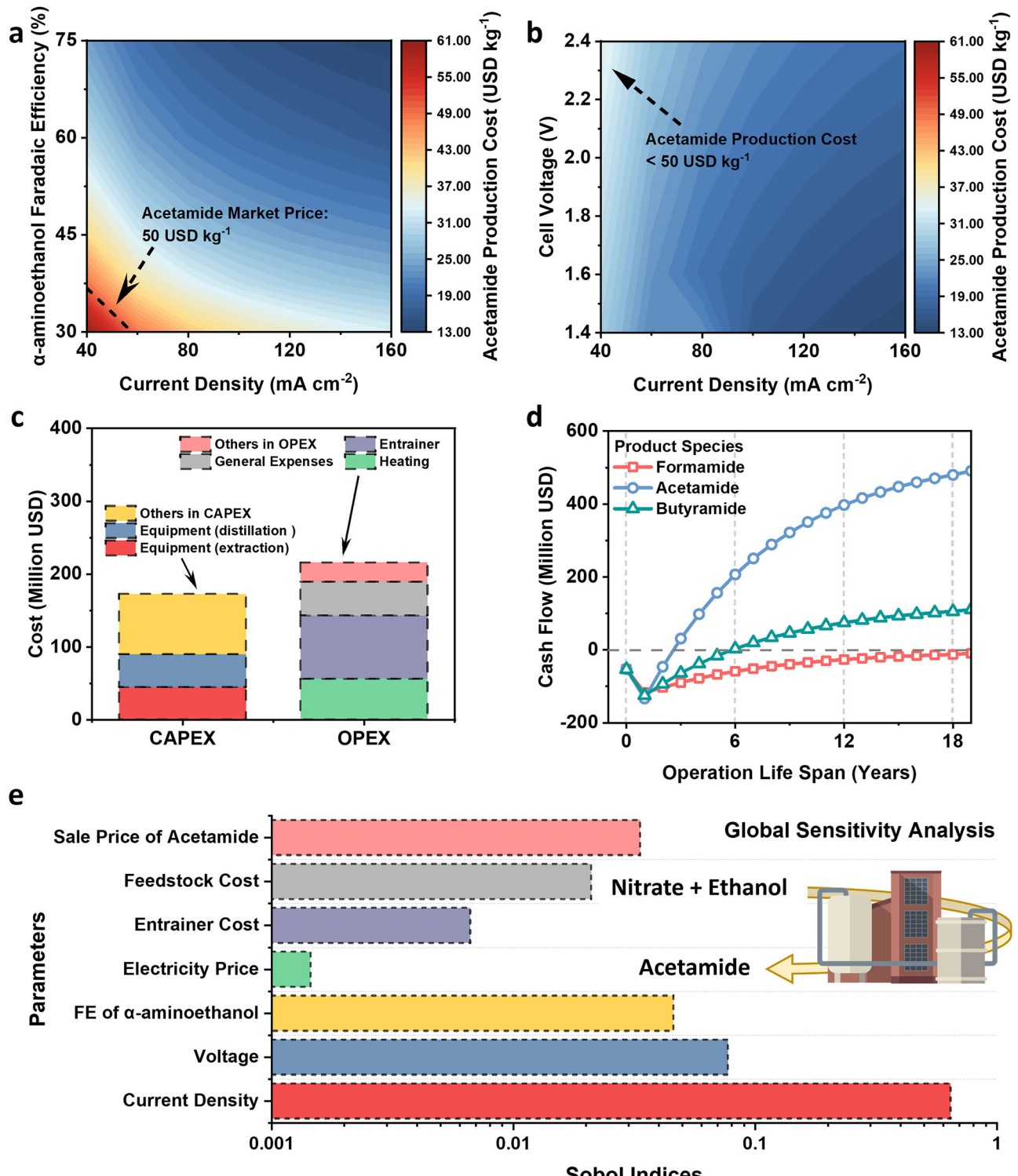

**Fig. 5 | Technical and economic analyses of acetamide electrosynthesis.** Contour plots of levelized cost as a function of **a** the current density and α-aminoethanol Faraday efficiency at 2.4 V cell voltage and **b** the current density and cell voltage at 52.5% α-aminoethanol Faraday efficiency; **c** Capital expenditures (million USD) and operating expenditures (million USD per year) for acetamide production. Others in CAPEX refer to equipments (reactor, ammonia stripper, cooler, pressure swing adsorption, pump, etc.). Others in OPEX refer to electricity, cooling, operation (labor related), maintenance, operating overhead, property taxes, and insurance and depreciation; **d** Cash flow charts for formamide, acetamide and butyramide production under 2.4 V cell voltage; **e** Global sensitivity analysis of various parameters for acetamide production.

## Material characterizations

The following equipment were used: STEM-HAADF (200 kV, FEI Titan Themis 60–300), EDS (200 kV, Super-X EDS system), TEM (200 kV, FEI Titan), SEM (JEOL JSM-6701F), XPS (Thermo Fisher Scientific K-Alpha+, monochromatic Al Kα), NMR (Bruker Advance 500 MHz), UV-Vis (Shimadzu UV-3600), ICP-OES (Perkin Elmer Avio 500, ppm-level accuracy), and Electrochemistry (Autolab PGSTAT30). For Rh K-edge XAS, 80 mg of 0.4 wt% $Rh_1/Ni(OH)_2$ powder (without Ni foam) was finely ground using a mortar and pestle, then pressed into a 5 mm pellet. Rh K-edge XAS spectra were collected at beamline 12-

BM-B of the Advanced Photon Source at Argonne National Laboratory using a double-crystal monochromator. Due to the low concentration of the element of interest in the sample, data were collected in fluorescence mode with a focused beam. Data analysis and simulations were performed using Athena, Artemis, and Hephaestus (Version 0.9.23).

## Electrochemically active surface area measurements

Electrochemically active surface area (ECSA) was measured through cyclic voltammetry (CV) from 1.078 to 1.130 V vs. RHE (the potential window without Faradic current densities) at different scan rates. The capacitance $\Delta j$ ($0.5 \ast |j_{charge} - j_{discharge}|$) was plotted as a function of scan rate with slope equal to the electrochemical double-layer capacitance ($C_{dl}$).

## Electrochemical ethanol oxidation in a H-cell setup

Experiments were carried out in a custom gas-tight H-type glass cell, separated by an anion exchange membrane (SELEMION AMVN), at room temperature (-24 °C) using an Autolab PGSTAT30 workstation. The three-electrode setup consisted of a saturated KCl Ag/AgCl reference electrode, a dimensionally stable anode (DSA, RuIr/TiO$_2$ on Ti-mesh, $1 \times 2$ cm$^2$), and a catalyst-modified Ni foam electrode ($1 \times 2$ cm$^2$). The aqueous electrolyte contained 1 M KOH and 1 M ethanol, with fresh solutions prepared immediately before use. Pre-prepared solutions were stored refrigerated (at 4 °C) until needed. Chronoamperometry (CA) was performed at a constant potential, and acetate concentration was measured by $^1$H NMR. The reported current density was normalized to the geometric area of nickel foam ($1 \times 1$ cm$^2$) without iR compensation. The tests in this part were conducted once due to the extended duration of the experiment.

## Electrochemical nitrate reduction coupling ethanol oxidation reaction in parallel in a flow electrolyzer setup

Measurements were conducted in a 16 cm$^2$ flow electrolyzer, separated by an anion exchange membrane (SELEMION AMVN), at room temperature (-24 °C) using an Autolab workstation with a maximum current of 10 A. The electrolyzer was assembled with two stainless-steel cover plates, two gold-coated copper current collectors, two monopolar graphite plates for electrolyte distribution, and two gaskets for the cathode and anode electrodes. Electrolytes for the cathode (1 M KOH and 1 M KNO$_3$) and anode (1 M KOH and 1 M ethanol) flowed into the respective electrodes without further circulation. Electrolyte solutions were freshly prepared prior to use, with any prepared solutions stored under refrigeration (e.g., at 4 °C) until needed. The flow rate was controlled by two Watson Marlow 120S peristaltic pumps, and the actual flow rate was verified using a measuring cylinder. Rh$_1$/Ni(OH)$_2$ on Ni foam ($4 \times 4$ cm$^2$) and Ru/Cu$_2$O on Cu foam ($4 \times 4$ cm$^2$) were used as the electrodes in a two-electrode setup. Chronoamperometry (CA) was performed at a constant potential. No iR correction was applied to the electrochemical data presented in this study. The tests in this part were conducted once due to the extended duration of the experiment.

## Electrochemical acetamide synthesis by coupling nitrate reduction reaction and ethanol oxidation reaction in series in a flow-electrolyzer setup

Measurements were conducted in a 16 cm$^2$ flow electrolyzer separated by an anion exchange membrane (SELEMION AMVN) at room temperature (-24 °C) using an Autolab workstation with a maximum current of 10 A. The electrolyzer setup included two stainless-steel cover plates, two gold-coated copper current collectors, two monopolar graphite plates with parallel paths for electrolyte distribution, and two gaskets for the cathode and anode electrodes. The electrolyte (0.1 M KOH, 1 M KNO$_3$, and 1 M ethanol) first flowed through the cathode, then to the anode without recirculation. Electrolyte solutions were prepared fresh immediately before use, with any pre-prepared solutions stored under refrigeration (e.g., at 4 °C) until needed. The flow rate was controlled by a Watson Marlow 120S peristaltic pump, and the actual flow rate was measured using a graduated cylinder. Rh$_1$/Ni(OH)$_2$ on Ni foam ($4 \times 4$ cm$^2$) and Ru/Cu$_2$O on Cu foam ($4 \times 4$ cm$^2$) were used as the electrodes in a two-electrode system. Chronoamperometry (CA) was performed at a constant potential. The flow-out solution from the catholyte chamber was collected in a glass vial and bubbled with CO$_2$ gas for 1 min to neutralize the solution. Stability tests were conducted at a flow rate of 1.5 mL min$^{-1}$ for continuous operation over 50 h at a full cell voltage of 2.4 V. No iR correction was applied to the electrochemical data presented in this study. The tests in this part were conducted three times to obtain error bars, but the long-term stability test was conducted once due to the extended duration of the experiment.

## $^1$H NMR determination of formate

The concentration of formate was quantified using $^1$H nuclear magnetic resonance ($^1$H NMR, 500 MHz). The internal standard was prepared by diluting 10 μL of DMSO 100-fold with water, followed by mixing with D$_2$O at a 1:1 (v/v) ratio. A calibration curve was generated using a series of standard sodium formate solutions. For each calibration point, 0.6 mL of the solution was mixed with 0.1 mL of the internal standard and analyzed using a 500 MHz NMR spectrometer. The calibration curve was constructed from the peak area ratio of formate to DMSO. For actual samples, the electrolyte was neutralized by bubbling CO$_2$ prior to NMR analysis, and measurements were performed using the same procedure.

## $^1$H NMR determination of acetate

The concentration of acetate was quantified using $^1$H nuclear magnetic resonance ($^1$H NMR, 500 MHz). The internal standard was prepared by diluting 10 μL of DMSO 100-fold with water, followed by mixing with D$_2$O at a 1:1 (v/v) ratio. A calibration curve was generated using a series of standard sodium acetate solutions. For each calibration point, 0.6 mL of the solution was mixed with 0.1 mL of the internal standard and analyzed using a 500 MHz NMR spectrometer. The calibration curve was constructed from the peak area ratio of acetate to DMSO. For actual samples, the electrolyte was neutralized by bubbling CO$_2$ prior to NMR analysis, and measurements were performed using the same procedure.

## $^1$H NMR determination of butyrate

The concentration of butyrate was quantified using $^1$H nuclear magnetic resonance ($^1$H NMR, 500 MHz). The internal standard was prepared by diluting 10 μL of DMSO 100-fold with water, followed by mixing with D$_2$O at a 1:1 (v/v) ratio. A calibration curve was generated using a series of standard sodium butyrate solutions. For each calibration point, 0.6 mL of the solution was mixed with 0.1 mL of the internal standard and analyzed using a 500 MHz NMR spectrometer. The calibration curve was constructed from the peak area ratio of butyrate to DMSO. For actual samples, the electrolyte was neutralized by bubbling CO$_2$ prior to NMR analysis, and measurements were performed using the same procedure.

## $^1$H NMR determination of formamide

The concentration of formamide was quantified using $^1$H nuclear magnetic resonance ($^1$H NMR, 500 MHz). The internal standard was prepared by diluting 10 μL of DMSO 100-fold with water, followed by mixing with D$_2$O at a 1:1 (v/v) ratio. A calibration curve was generated using a series of standard formamide solutions. For each calibration

point, 0.6 mL of the solution was mixed with 0.1 mL of the internal standard and analyzed using a 500 MHz NMR spectrometer. The calibration curve was constructed from the peak area ratio of formamide to DMSO. For actual samples, the electrolyte was neutralized by bubbling $CO_2$ prior to NMR analysis, and measurements were performed using the same procedure.

### $^1$H NMR determination of acetamide

The concentration of acetamide was quantified using $^1$H nuclear magnetic resonance ($^1$H NMR, 500 MHz). The internal standard was prepared by diluting 10 μL of DMSO 100-fold with water, followed by mixing with $D_2O$ at a 1:1 (v/v) ratio. A calibration curve was generated using a series of standard acetamide solutions. For each calibration point, 0.6 mL of the solution was mixed with 0.1 mL of the internal standard and analyzed using a 500 MHz NMR spectrometer. The calibration curve was constructed from the peak area ratio of acetamide to DMSO. For actual samples, the electrolyte was neutralized by bubbling $CO_2$ prior to NMR analysis, and measurements were performed using the same procedure.

### $^1$H NMR determination of butyramide

The concentration of butyramide was quantified using $^1$H nuclear magnetic resonance ($^1$H NMR, 500 MHz). The internal standard was prepared by diluting 10 μL of DMSO 100-fold with water, followed by mixing with $D_2O$ at a 1:1 (v/v) ratio. A calibration curve was generated using a series of standard butyramide solutions. For each calibration point, 0.6 mL of the solution was mixed with 0.1 mL of the internal standard and analyzed using a 500 MHz NMR spectrometer. The calibration curve was constructed from the peak area ratio of butyramide to DMSO. For actual samples, the electrolyte was neutralized by bubbling $CO_2$ prior to NMR analysis, and measurements were performed using the same procedure.

### UV-Vis determination of ammonium

The concentration of ammonium was determined using a modified indophenol blue method, measured by UV-Vis spectroscopy[32,33]. A specific volume of electrolyte was collected and diluted to fall within the detection range (typically 50 to 200 times). To each 2 mL of the diluted sample, 2 mL of a solution containing 5 wt% salicylic acid and 5 wt% sodium citrate in 1 M NaOH was added, followed by 1 mL of 0.05 M sodium hypochlorite and 0.2 mL of 1 wt% sodium nitroferricyanide solution. After standing for 2 h at room temperature, the indophenol blue formation was analyzed by UV-Vis spectroscopy, with absorbance measured at the maximum wavelength of 656 nm. Ammonium concentration was then calculated using an external calibration curve prepared from standard ammonium chloride solutions ranging from 0 to 0.5 mM.

### UV-Vis determination of nitrite

The concentration of nitrite was determined using a modified diazotization colorimetry method, measured by UV-Vis spectroscopy[10]. A specific volume of electrolyte was collected and diluted to fall within the detection range (typically 100 to 2500 times dilution). To 5 mL of the diluted solution, 0.1 mL of a reagent mixture containing 40 g L$^{-1}$ p-aminobenzenesulfonamide, 100 mL/L phosphoric acid, and 2 g L$^{-1}$ N-(1-naphthyl)-ethylenediamine dihydrochloride was added. After standing for 20 min at room temperature, the solution was analyzed by UV-Vis spectroscopy, with absorbance measured at the maximum wavelength of 540 nm. Nitrite concentration was then calculated using an external calibration curve created from standard potassium nitrite solutions ranging from 0 to 0.08 mM.

### UV-Vis determination of nitrate

The concentration of nitrate was determined using a modified UV-Vis spectroscopy method[34]. A specific volume of electrolyte was collected

and diluted to the detection range (typically 100 to 10,000 times dilution). To 10 mL of the diluted solution, 0.2 mL of 4 M HCl and 0.02 mL of a 0.8 wt% sulfamic acid solution were added. After standing for 10 min at room temperature, the solution was analyzed by UV-Vis spectroscopy, with absorbance measured at 220 and 275 nm. The standard curve was constructed using the value of $A_{220nm} - 2A_{275nm}$ ($A$ is the absorbance intensity) plotted against nitrate concentration. Nitrate concentration was determined using an external calibration curve prepared from standard potassium nitrate solutions ranging from 0 to 0.5 mM.

### $^1$H NMR validation of butyrate generation from the cathode in a flow electrolyzer

$^1$H NMR measurements were conducted using an AVNEO500 NMR spectrometer. The electrolyte consisted of a water solution containing 0.1 M KOH, 1 M $KNO_3$, and 1 M tetrahydrofuran. The flow electrolyzer operated at 60 °C with an electrolyte flow rate of 1.5 mL min$^{-1}$, applying a voltage of 2.4 V. Before NMR analysis, the pH of the post-reaction electrolyte was neutralized by bubbling $CO_2$ through it. For NMR analysis, 0.6 mL of the reacted electrolyte was mixed with 0.1 mL of the internal standard and then tested in the 500 MHz NMR spectrometer.

### $^1$H NMR validation of 4-cyanobenzenediazonium generation from the cathode in a flow electrolyzer

$^1$H NMR measurements were performed using an AVNEO500 NMR spectrometer. The electrolyte consisted of a water solution containing 0.01 M HCl, 1 M $KNO_3$, and 1 M 4-aminobenzonitrile. The flow electrolyzer operated at room temperature with an electrolyte flow rate of 1.5 mL min$^{-1}$, applying a voltage of 2.4 V. After the reaction, 0.6 mL of the electrolyte was mixed with 0.1 mL of the internal standard and analyzed using the 500 MHz NMR spectrometer.

### $^1$H NMR validation of α-aminoethanol generation from the cathode in a flow electrolyzer

$^1$H NMR measurements were conducted using an AVNEO500 NMR spectrometer. The electrolyte consisted of a water solution containing 0.1 M KOH, 1 M $KNO_3$, and 1 M ethanol ($CH_3CH_2OH$). The flow electrolyzer operated at room temperature with an electrolyte flow rate of 1.5 mL min$^{-1}$, applying a voltage of 2.4 V. The pH of the post-reaction electrolyte was neutralized by bubbling $CO_2$ through it prior to NMR analysis. A 0.6 mL sample of the reacted electrolyte was mixed with 0.1 mL of the internal standard and analyzed using the 500 MHz NMR spectrometer.

### $^1$H NMR validation of reactants required for α-aminoethanol

$^1$H NMR measurements were conducted using an AVNEO500 NMR spectrometer. The reaction was carried out in a 20 mL water solution containing 0.1 M KOH, 0.1 M $KNO_2$, 0.05 M $(NH_4)_2CO_3$, and 1 M ethanol ($CH_3CH_2OH$) in a tube. The solution was allowed to stand for 2 min before data acquisition. Prior to NMR analysis, the pH of the post-reaction electrolyte was neutralized by bubbling $CO_2$ through it. A 0.6 mL sample of the electrolyte was then mixed with 0.1 mL of the internal standard and analyzed using the 500 MHz NMR spectrometer.

### Differential electrochemical mass spectrometer validation of the NO intermediate

DEMS was conducted with QAS100 in Linglu Instruments (Shanghai) Co., Ltd. 50 μm thickness water-resistant and breathable PTFE was employed as membrane with ≥50% porosity and ≤20 nm pore size. The reaction used Ru/$Cu_2O$ for $NO_3RR$ in an electrolyte of 0.1 M KOH, 0.1 M $KNO_3$, and 0.1 M $CH_3CH_2OH$ at 25 mA cm$^{-2}$. Each sample was tested for 15 min for data acquisition.

## Products calculation

Acetamide synthesis:

$$CH_3CH_2OH + NO_3^- + 4H_2O + 6e^- \rightarrow CH_3CH(OH)NH_2 + 7OH^- \text{ (Cathode)}$$

$$CH_3CH(OH)NH_2 + 2OH^- - 2e^- \rightarrow CH_3CONH_2 + 2H_2O \text{(Anode)}$$

Side reactions:

$$NO_3^- + 6H_2O + 8e^- \rightarrow NH_3 + 9OH^- \text{ (Cathode)}$$

$$CH_3CH_2OH + 4OH^- - 4e^- \rightarrow CH_3COOH + 3H_2O \text{ (Anode)}$$

Therefore, in our system, one acetamide, ammonia, and acetate produced corresponds to eight, eight, and four electron transfers, separately. Assuming that the acetamide and acetate concentration obtained by NMR is $c_{Acetamide}$ (mM) and $c_{Acetate}$ (mM), separately. Ammonia concentration obtained by UV-Vis is $c_{Ammonia}$ (mM). The current of flow electrolyzer is $i$ (A), and flow rate is $v$ (mL min$^{-1}$). Faraday constant is $F$ ($9.6485332 \times 10^4$ C mol$^{-1}$)

$$FE_{Acetamide} = c_{Acetamide} \times v \times 8e^- \times F \\ \times (i \times 60\,s \times 1000 \times 1000)^{-1} \times (100\%) \tag{1}$$

$$\text{Yield Rate}_{Acetamide}\left(mmol\,h^{-1}\right) = c_{Acetamide} \times v \times 3600\,s \times (60\,s \times 1000)^{-1} \tag{2}$$

$$FE_{Ammonia} = c_{Ammonia} \times v \times 8e^- \times F \times (i \times 60\,s \times 1000 \times 1000)^{-1} \times (100\%) \tag{3}$$

$$\text{Yield Rate}_{Ammonia}\left(mmol\,h^{-1}\right) = c_{Ammonia} \times v \times 3600\,s \times (60\,s \times 1000)^{-1} \tag{4}$$

$$FE_{Acetate} = c_{Acetate} \times v \times 4e^- \times F \times (i \times 60\,s \times 1000 \times 1000)^{-1} \times (100\%) \tag{5}$$

$$\text{Yield Rate}_{Acetate}\left(mmol\,h^{-1}\right) = c_{Acetate} \times v \times 3600\,s \times (60\,s \times 1000)^{-1} \tag{6}$$

## DFT computations

Density functional theory (DFT) calculations were performed using the projector augmented wave (PAW) basis[35] and the revised Perdew–Burke–Ernzerhof (PBE) exchange-correlation functional[36], as implemented in the Vienna Ab Initio Simulation Package (VASP 5.4.4.18)[37,38]. The plane-wave cutoff energy was set to 400 eV. Brillouin zone integration was carried out using $\Gamma$-centered $2 \times 2 \times 1$ k-point grids for the adsorption energy calculations. Convergence criteria for total energy and atomic forces were set to less than $10^{-5}$ eV and 0.03 eV Å$^{-1}$, respectively.

## Model development for techno-economic analysis

Two process models in this study were developed using the Aspen Plus V14 modeling environment, based on experimental data. The non-random two-liquid (ELECNRTL) model was employed as the physical property package for all blocks except the extraction block, which was simulated using the NRTL model. Two RStio reactors were used to represent the anode and cathode in the electrolyzer, while RadFrac blocks were utilized for all distillation processes. Ethyl acetate was used as an entrainer to aid in the separation of acetate from the

electrolyte and to facilitate the acetamide purification process, eliminating residual water. Low-pressure ammonia was separated by the stripper (see Table S3). The power across all cases was kept constant at 10 MW, which can be generated by a single PV farm (40 MW with 20% efficiency).

## Parameters for techno-economic analysis

The procedure for the techno-economic evaluation, based on the values obtained from the nitrate reduction/hydrogen oxidation case study simulation, is outlined in Table S4. The base framework was adapted from the works of Jonggeol Na et al.[39] and seider et al.[40].

## Cash flow and levelized cost

Cash flow and levelized cost analysis were performed assuming a 20-year plant life ($N_{life}$), a 1-year plant construction period, straight-line depreciation, a 5% nominal interest rate, and a 22% income tax rate to calculate the net present value (NPV), using the equations below:

$$\text{Net Earning} = (\text{Sell} - \text{Depreciation} - C_{Excl.Dep.}) \cdot (100\% - tax_{income})$$

$$\text{Annual Cash Flow} = (\text{Net Earning} + C_D) \\ - Total\ capital\ invesment(Depreciable)$$

$$NPV = \sum_{n=1}^{N_{life}} \frac{CF_n}{(1+i)^n}$$

$$\text{Net Present Value}(\text{Levelized Cost}_{Ammonia}) = 0$$

The production cost, excluding depreciation ($C_{Excl.Dep.}$), included expenses for feedstocks (nitrate wastewater, electrolyte, and $H_2$), utilities, labor, maintenance, operating overhead, property taxes, insurance, and general expenses. The levelized cost of ammonia was calculated by setting the NPV equal to zero. The upper and lower bounds for the levelized cost are shown in Table S5.

## Data availability

All data were available from the authors upon request. Source data are provided with this paper.

## Code availability

All code related to this paper may be requested from the authors.

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

## Acknowledgements

K.P.L. acknowledges funding support from NUS's Centre for Hydrogen Innovation program CHI-P2022-01.

## Author contributions

Q.H. conceived the research, synthesized the materials, and conducted catalytic measurements under the supervision of K.P.L.; Technical and economic analyses were conducted by O.P.; In situ XAS and XRD were performed by J.L.; STEM-HAADF and HRTEM were performed by M.S.; XPS measurements were performed by J.F. and Z.X.; SEM was performed by K.Z.; DFT calculations were performed by Q.H.; Large current flow cell operation was performed with the assistance of J.L. and D.C.; The draft was written by Q.H., revised by K.P.L. All authors discussed and commented on the manuscript.

## Competing interests

The authors declare no competing interests.
