## [Transparent Peer Review file · Nature Communications]

Coupling Nitrate Electrochemical Reduction and Nitrite Oxidation of Ethanol for Acetamide Synthesis

Corresponding Author: Professor Kian Ping Loh

Version 0:

Reviewer comments:

Reviewer #1

(Remarks to the Author)

This manuscript proposes an innovative electrochemical approach for acetamide synthesis, which also shows potential applicability to formamide and butyramide synthesis. DFT and technical and economic analyses demonstrate the feasibility of this new acetamide electrosynthesis route. The research topic is worth investigating and the data are comprehensive. Therefore, this work may be considered for publication. However, several critical concerns require clarification:

1. The article describes a reaction pathway where ethanol reacts with nitrite at the cathode to form acetaldehyde, which subsequently reacts with NH_3 to generate α -aminoethanol. Why is acetaldehyde not reduced at the cathode? The presence of an oxidation reaction at the cathode is puzzling and intriguing.
2. The article notes that ethanol oxidation to acetic acid at the anode is a competing pathway if the reaction initiates there. However, critical experimental details are missing: 1. What was the initial composition of the anolyte; 2. Was the catholyte circulated to the anode compartment immediately, or was this delayed until cathodic products (e.g., ammonia) reached a threshold concentration?
3. The article emphasizes selecting catalysts with EOR performance at the anode. However, the EOR is a competing reaction and the primary objective of the anode should be to promote the oxidation of α -aminoethanol.
4. The figure labeling in the manuscript appears disordered and inconsistent with the chronological sequence of content description (e.g., Figure 4c). Additionally, many figures in the Supporting Information (SI) are not referenced in the main text.
5. Based on Figure S26, the proton signal of acetamide overlaps with the solvent peak, rendering the quantitative analysis via ^1H NMR spectroscopy unreliable. Other more suitable solvents should be used.
6. For qualitative analysis of the products via ^1H NMR spectroscopy, integration of proton signals or comparison with authentic reference standards is essential to confirm compound identity. Additionally, complementary analytical techniques such as LC-MS or GC-MS could be employed to further confirm the structural assignments.
7. The authors must provide detailed calculations for the Faradaic efficiencies of both electrodes (e.g., cathode Faradaic efficiency and anode Faradaic efficiency, as mentioned in the article).
8. Is there the structural reconstruction of $\text{Rh}_1/\text{Ni}(\text{OH})_2$ during oxidation process. Detailed in situ- and post-reaction characterizations (e.g., XPS, XRD, TEM) are needed to elucidate the catalyst's reconstruction/transformation and identify the true active species responsible for catalytic activity.

Reviewer #2

(Remarks to the Author)

This manuscript presents a method for the electrosynthesis of amides. The nitrite produced at the cathode oxidizes ethanol to produce acetaldehyde, which then reacts with the ammonia produced at the cathode to form α -aminoethanol. Finally, α -aminoethanol is converted to acetamide by anodic oxidation. The authors suggest that the current reported acetamide Faradaic efficiency (<40%) and yield (<0.2 $\text{mmol h}^{-1} \text{cm}^{-2}$) are low, however, the result of this paper shows that Faradaic efficiency of acetamide is only 22%, and the yield is about 0.45 $\text{mmol h}^{-1} \text{cm}^{-2}$. Moreover, due to the complexity of the reaction system, authors should conduct more in-depth research. Therefore I do not recommend its publication.

1. Why are acetate products formed when methanol is used as a substrate in Figure S8 and Figure S9?
2. The authors suggest that HNO is not only a reduction product in ethanol oxidation reaction, but also an important

intermediate for ammonia production. Were the authors able to demonstrate the presence of HNO intermediate by spectroscopic or other characterization experiments?

3. Since the test was carried out in the interval where there is no Faraday reaction, what is the cause of the current fluctuation in Figure S5?

4. What is the Faraday side reaction of the cathode at low potential? Is there a reduction of acetaldehyde to ethanol?

5. The reaction system proposed by the authors requires the product produced by the cathode to enter the anode reaction. Does the rate of production of cathode products match the demand for anode? Is the insufficient supply of reactants the reason for the low Faraday efficiency (22%) of anode target reaction?

6. Could the authors quantify the α -aminoethanol produced by the cathode? Did the authors try to investigate anode reaction during α -aminoethanol as sole reactant?

7. Will the ammonia produced by the cathode be oxidized when it enters the anode?

8. Both 1 M KOH and 0.1 M KOH were used in the study. How does the concentration of KOH affect the reaction? Is acetamide stable in alkaline electrolytes?

Reviewer #3

(Remarks to the Author)

The authors reported that a new synthesis avenue to prepare acetamide by coupling nitrate electrochemical reduction and nitrite oxidation of ethanol. The new insights reported in this manuscript are significant and attractive, and may be further improved by focusing on or putting more emphasis on analysis of catalyst structure and product after continuous acetamide synthesis. Some revisions are required before the final publication:

1. To examine the structural integrity of the electrocatalyst, it would be benefited to conduct further structural analysis of Rh1/Ni(OH)₂ catalyst before and after continuous acetamide synthesis at 2.4 V cell voltage for 3000 minutes (Figure 4e). Has Rh single atoms aggregated? Has the Ni(OH)₂ support corroded?

2. What are the by-products in the continuous acetamide synthesis? How was the purified acetamide obtained?

3. The selectivity of nitrate reduction to nitrite in the Ru/Cu₂O cathode has not been quantified, which may affect the reaction efficiency.

4. Figure 2e does not have any corresponding explanations in the manuscript.

Version 1:

Reviewer comments:

Reviewer #1

(Remarks to the Author)

The author has addressed my questions well. However, I still have a minor concern, please refer to the following questions.

The experimental description presents some apparent contradictions regarding ethanol introduction timing and reaction initiation control. In Page 2, the authors stated that "Ethanol was added to the cathode during nitrate reduction, where it was oxidized to acetaldehyde by nitrite, an intermediate in nitrate reduction." However, Page 15 clarifies that the electrolyte (containing 0.1 M KOH, 1 M KNO₃ and 1 M CH₃CH₂OH) flows into the cathode first, then proceeds to the anode without further circulation. This suggests ethanol was present from the beginning rather than being added mid-reaction. Page 3 further notes: "We found that if the reaction starts from the anode, the production of acetate competes with acetaldehyde, resulting in a much poorer acetamide yield rate." This raises important questions about: (1) the precise mechanism ensuring cathode-first reaction initiation (as opposed to anode initiation), and (2) what reactions occur at the anode during the initial phase. The authors' repeated emphasis on the flow cell's special environment would be better supported by including a schematic diagram or a video to clarify these operational details.

Reviewer #2

(Remarks to the Author)

Authors have addressed reviewers' concerns largely. It could be accepted.

Version 2:

Reviewer comments:

Reviewer #1

(Remarks to the Author)

The authors have well addressed the questions I raised, and I recommend accepting this manuscript.

Point-by-point responses to reviewers' comments for "Coupling Nitrate Electrochemical Reduction and Nitrite Oxidation of Ethanol for Acetamide Synthesis" (NCOMMS-25-17318-T)

Reviewer #1:

*This manuscript proposes an **innovative** electrochemical approach for acetamide synthesis, which also shows potential applicability to formamide and butyramide synthesis. DFT and technical and economic analyses demonstrate the feasibility of this new acetamide electrosynthesis route. **The research topic is worth investigating and the data are comprehensive.** Therefore, this work may be considered for publication. However, several critical concerns require clarification:*

Response: We sincerely thank the reviewer for his/her positive comments on our manuscript. We have supplemented experimental details, product validation and catalyst stability characterizations in manuscript and supporting information. Please find our point-by-point response below.

Question 1: *The article describes a reaction pathway where ethanol reacts with nitrite at the cathode to form acetaldehyde, which subsequently reacts with NH₃ to generate α -aminoethanol. Why is acetaldehyde not reduced at the cathode? The presence of an oxidation reaction at the cathodic is puzzling and intriguing.*

Response: Thank you for raising this thought-provoking question. Acetaldehyde's absence of reduction at the cathode can be attributed to its inherent instability in aqueous environments. Our NMR analysis (Figure R1) of a freshly prepared 50 mM acetaldehyde solution (acetaldehyde purchased from Sigma-Aldrich) revealed rapid decomposition into multiple side products within minutes (refs: 10.1002/0471264180.or016.01, 10.1021/ie5004043, 10.1002/aic.14623). To circumvent these side reactions, we designed a system where ethanol is in situ oxidized to acetaldehyde via nitrite at the cathode. This strategy ensures that acetaldehyde reacts immediately with NH₃ to form α -aminoethanol, minimizing its accumulation and potential degradation.

The observed oxidation of ethanol at the cathode is enabled by the unique electrochemical environment in the flow cell. Here, concentrated nitrite not only drives ethanol oxidation but also facilitates the coupling of acetaldehyde with NH₃, a process unachievable in conventional H-cell setups (as evidenced by the absence of α -aminoethanol in Figure R2). This synergy between nitrite-mediated oxidation and rapid aldehyde-ammonia reaction underscores the necessity of the flow cell for efficient product generation.

In summary, the cathode's role in this pathway is twofold: nitrite acts as an oxidizing agent for ethanol, while the flow configuration ensures rapid consumption of acetaldehyde, thereby bypassing its reduction or decomposition. This approach highlights the importance of tailored electrochemical environments in steering reaction pathways.

Figure R1. ^1H NMR validation of acetaldehyde stability in water solution. Many side products can be observed.

Figure R2. ^1H NMR validation of nitrate reduction with ethanol in H-cell. The electrolyte was the same as we used in flow electrolyzer (0.1 M KOH, 1 M KNO_3 and 1 M $\text{CH}_3\text{CH}_2\text{OH}$). The reaction was run at 200 mA for 5 minutes. No α -aminoethanol was observed.

Question 2: *The article notes that ethanol oxidation to acetic acid at the anode is a competing pathway if the reaction initiates there. However, critical experimental details are missing: 1. What was the initial composition of the anolyte; 2. Was the catholyte circulated to the anode compartment immediately, or was this delayed until cathodic products (e.g., ammonia) reached a threshold concentration?*

Response: Thank you for the comment. We have updated the manuscript and supporting information to include the electrolyte composition. Regardless of whether the reaction starts at the anode or cathode, the electrolyte remains the same, without circulation.

For Question 2.1: The initial analyte composition is 0.1 M KOH, 1 M KNO₃, and 1 M CH₃CH₂OH.

For Question 2.2: The catholyte was not circulated to the anode compartment. The reaction begins at the anode and ends at the cathode without circulation.

This underscores the significance of our pathway design. If the reaction starts at the anode, the generated acetaldehyde is highly unstable (as shown in Question 1). Acetaldehyde can rapidly undergo other side reactions at the anode. While α -aminoethanol is also not highly stable, it is significantly more stable than acetaldehyde in water solution, allowing its clear signal to be observed in ¹H NMR. Once α -aminoethanol is generated at the cathode, it is transferred to the anode immediately to maximize acetamide yield rate.

Question 3: *The article emphasizes selecting catalysts with EOR performance at the anode. However, the EOR is a competing reaction and the primary objective of the anode should be to promote the oxidation of α -aminoethanol.*

Response: Thank you for this critical comment. We selected ethanol oxidation to quantitatively assess the oxidation ability of our catalysts. While performing α -aminoethanol oxidation at the anode would be ideal, α -aminoethanol is an unstable (**Figure R3**) and uncommon compound, so it is not commercially available from major reagent suppliers (Sigma-Aldrich, Tokyo Chemical Industry, and BLDpharm).

Given that our final product, acetamide, has the same carbon valence state as acetate, we use ethanol oxidation as a proxy to evaluate catalyst performance in acetamide synthesis. Additionally, we tested acetamide synthesis using bulk Ni(OH)₂ as the anode in our system, which resulted in a lower yield rate (5.0 mmol h⁻¹, **Figure R4**).

Figure R3. ^1H NMR validation of α -aminoethanol stability. ^1H NMR spectra of (a) fresh α -aminoethanol in cathode electrolyte from the 16 cm^2 flow electrolyzer under 2.4 V cell voltage and (b) reduced α -aminoethanol in the same electrolyte after 20 min. The peak area of α -aminoethanol decreased from 0.4768 to 0.3587 vs. DMSO, indicating some α -aminoethanol decomposed.

Figure R4. Acetamide synthesis performance by using (a) Rh₁/Ni(OH)₂ and (b) Ni(OH)₂ as anode.

Question 4: *The figure labeling in the manuscript appears disordered and inconsistent with the chronological sequence of content description (e.g., Figure 4c). Additionally, many figures in the Supporting Information (SI) are not referenced in the main text.*

Response: Thank you for this critical comment. We have revised figure order to make them follow the sequence of content description. We have also added description in manuscript for all figures in the Supporting Information. You can refer to revision part and revised manuscript.

Question 5: *Based on Figure S26, the proton signal of acetamide overlaps with the solvent peak, rendering the quantitative analysis via ¹H NMR spectroscopy unreliable. Other more suitable solvents should be used.*

Response: Actually, the interfering peak is from the acetaldehyde and acetate product, not the solvent, and this overlap is unavoidable. To address the quantification issue, we applied MestReNova software to fit the peaks of acetamide, acetaldehyde and acetate with lowest residue, and then integrate them (<https://mestrelab.com/wp-content/uploads/2021/09/mnova-2024-05-23-mnova-9.pdf>, 8.9 Line Fitting (Deconvolution), Page 322).

Besides, we weighed the purified acetamide powder from 45 mL electrolyte after stability test (corresponding to synthesized product in 30 minutes), obtaining 0.1953 g. Given the molecular weight of acetamide (59.07 g/mol), this corresponds to a production rate of 6.6 mmol h⁻¹. Apart from potential product loss during the purification process, this result is generally consistent with the value determined by NMR quantification.

Revised insert figure in Figure 4e. Purified acetamide (0.1953 g from 45 mL electrolyte) collected after stability test.

Question 6: *For qualitative analysis of the products via ^1H NMR spectroscopy, integration of proton signals or comparison with authentic reference standards is essential to confirm compound identity. Additionally, complementary analytical techniques such as LC-MS or GC-MS could be employed to further confirmed the structural assignments.*

Response: Thank you for the comment. We also applied GC-MS to examine our purified acetamide (50 mM in ethyl acetate). The results are shown as Figure S36. The discussion has been added in the manuscript:

“Gas chromatography-Mass spectrometry (GC-MS) confirmed structural integrity of synthesized acetamide (Figure S36).”

Figure S36. (a) Gas chromatography-Mass spectrometry (SHIMADZU GCMS-QP2020) for purified acetamide (50 mM in ethyl acetate); (b) Standard spectrum of acetamide in the database of GC-MS.

Question 7: The authors must provide detailed calculations for the Faradaic efficiencies of both electrodes (e.g., cathode Faradaic efficiency and anode Faradaic efficiency, as mentioned in the article).

Response: Thank you for this critical comment. We have added this part in experimental section of supporting information.

Question 8: Is there the structural reconstruction of Rh₁/Ni(OH)₂ during oxidation process. Detailed in situ- and post-reaction characterizations (e.g., XPS, XRD, TEM) are needed to elucidate the catalyst's reconstruction/transformation and identify the true active species responsible for catalytic activity.

Response: Thank you for this critical comment. Following the referee's comment, we have conducted in-situ XAS to demonstrate that the structure of Ni(OH)₂ remained unchanged. Therefore, there is no significant structure reconstruction. Additionally, XRD, XPS and XAS revealed no observable changes of both Rh and Ni in Rh₁/Ni(OH)₂ catalyst after the stability test. Furthermore, SEM confirmed that the morphology of Rh₁/Ni(OH)₂ catalyst also didn't change after the stability test.

In-situ XAS: We performed in-situ XAS to detect any changes in the coordination environment of the Rh₁/Ni(OH)₂ catalyst (Figure S38) during the reaction. In ethanol oxidation, Ni K-edge XANES in Rh₁/Ni(OH)₂ catalyst did not change at current density of 50 mA cm⁻² and 100 mA cm⁻² for 20 or 40 minutes.

Figure S38. Ni K-edge XANES spectra of (a) Ni foil and Rh₁/Ni(OH)₂ at different current density in ethanol oxidation; (b) Rh₁/Ni(OH)₂ at different current density and different duration time in ethanol oxidation. 1 M KOH and 1 M CH₃CH₂OH was used as electrolyte.

XRD, XPS and XAS: We have conducted XRD, XPS and XAS analyses to evaluate Rh₁/Ni(OH)₂ catalyst stability before and after the stability test. XRD pattern revealed that Ni(OH)₂ structure remained unchanged (Figure S39a). The Ni 2p XPS spectra indicated that the valence state of Ni remained unchanged (Figure S39b). Similarly, Rh K-edge EXAFS spectra confirmed that the coordination environment of Rh was preserved throughout the stability assessment (Figure S39c).

Figure S39. (a) XRD pattern of $\text{Rh}_1/\text{Ni}(\text{OH})_2$ catalyst, (b) XPS Ni_{2p} core-level spectra of $\text{Rh}_1/\text{Ni}(\text{OH})_2$ catalyst and (c) Rh K -edge EXAFS spectra of $\text{Rh}_1/\text{Ni}(\text{OH})_2$ catalyst before and after stability test.

SEM: Furthermore, SEM images confirmed that there is no change in the morphology of the $\text{Rh}_1/\text{Ni}(\text{OH})_2$ catalyst before and after the reaction (Figure S40).

Figure S40. SEM images of $\text{Rh}_1/\text{Ni}(\text{OH})_2$ catalyst on Ni foam after stability test at (a) 25000x and (b) 6000x magnification.

The discussion has been added in the manuscript:

“To evaluate the stability of the $\text{Rh}_1/\text{Ni}(\text{OH})_2$ catalyst, we performed in situ XAS (Figure S38). During ethanol oxidation, the Ni K -edge XANES spectra remained unchanged at current densities of 50 mA cm^{-2} and 100 mA cm^{-2} for 20 and 40 minutes, indicating that the $\text{Ni}(\text{OH})_2$ structure was preserved. Post-stability test analyses using XRD, XPS and XAS further confirmed that there is no change in coordination environment for the active sites in the catalyst. The XRD pattern showed no structural changes in $\text{Ni}(\text{OH})_2$ (Figure S39a), and Ni $2p$ XPS spectra indicated that the Ni valence state (Figure S39b) did not change. Rh K -edge EXAFS spectra demonstrated that Rh's coordination environment remained stable (Figure S39c). Additionally, SEM images showed no noticeable morphological changes at the end of the reaction (Figure S40).”

Revision:

1. We have supplemented the electrolyte composition in experimental section of manuscript and supporting information.
2. We have supplemented the calculation for Faradaic efficiency and yield rate of products in experimental

section of supporting information.

3. We have added the sentence in manuscript: "We have also examined electrochemical active surface area (ECSA) of Ni(OH)₂ and Rh₁/Ni(OH)₂ catalysts (Figure S5). The presence of Rh single atom slightly increases the active area of Ni(OH)₂." in Page 4.
4. We have added the sentence in manuscript: "The calibration curves of all the products tested are shown in Figures S6 and S7." in Page 4.
5. We have revised the sentence into "As shown in **Figure 2d**, this system achieves 9.4 A current at 2.4 V cell voltage, synthesizing ammonia at 90.7% Faradaic efficiency with 39.7 mmol h⁻¹ yield rate and acetate at 81.6% Faradaic efficiency 71.6 mmol h⁻¹ yield rate (**Figure 2e** and **2f**), which is significantly higher than using Ni(OH)₂ as anode (Figure S9)." in Page 5.
6. We have added the sentence in manuscript: "Lower Rh loading of 0.1 wt% and higher Rh loading of 1 wt% (Rh nanoparticles, Figure S11) on Ni(OH)₂ were also synthesized, showing poorer performance in ethanol oxidation (Figure S12)." in Page 5.
7. We have added the sentence in manuscript: "α-aminoethanol intermediate was detected in the cathode electrolyte during electrocatalytic process (Figure S24)." in Page 8.
8. We have revised the sentence into "Subsequently, the electro-synthesized acetamide was identified and quantified from anode electrolyte by ¹H NMR (Figure S25 and S26)." in Page 9.
9. We have added the sentence in manuscript: "In such system, direct nitrate reduction to ammonia and alcohol oxidation to carboxylate are the two main side reactions. The side products of ammonia and carboxylate are quantified in Figure S33." in Page 9.
10. We have added discussion for control experiments in manuscript: "Control experiments using bulk Ni(OH)₂ as anode (Figure S34a) shows that it has poorer acetamide synthesis performance than Rh₁/Ni(OH)₂. Using the same electrolyte but reversing the flow direction in a single pass (from anode and ends at cathode) decreased the acetamide yield significantly (Figure S34b), highlighting the crucial role of ethanol oxidation by nitrite in acetamide synthesis." in Page 9.
11. We have revised the sequence of **c** and **d** in **Figure 4** in page 10.
12. We have added the sentence in manuscript: "Gas chromatography-Mass spectrometry (GC-MS) confirmed structural integrity of synthesized acetamide (Figure S36)." in Page 11.
13. We have added the paragraph in manuscript: "To evaluate the stability of the Rh₁/Ni(OH)₂ catalyst, we performed in situ XAS (Figure S38). During ethanol oxidation, the Ni K-edge XANES spectra remained unchanged at current densities of 50 mA cm⁻² and 100 mA cm⁻² for 20 and 40 minutes, indicating that the Ni(OH)₂ structure was preserved. Post-stability test analyses using XRD, XPS and XAS further confirmed the that there is no change in coordination environment for the active sites in the catalyst. The XRD pattern showed no structural changes in Ni(OH)₂ (Figure S39a), and Ni 2p XPS spectra indicated that the Ni valence state (Figure S39b) did not change. Rh K-edge EXAFS spectra demonstrated that Rh's coordination environment remained stable (Figure S39c). Additionally, SEM images showed no noticeable morphological changes at the end of the reaction (Figure S40)." in Page 11.
14. We have added the sentence in manuscript: "The flowsheet of acetamide electrosynthesis is shown in Figure S41." in Page 11.

Reviewer #2:

This manuscript presents a method for the electrosynthesis of amides. The nitrite produced at the cathode oxidizes ethanol to produce acetaldehyde, which then reacts with the ammonia produced at the cathode to

form α -aminoethanol. Finally, α -aminoethanol is converted to acetamide by anodic oxidation. The authors suggests that the current reported acetamide Faradaic efficiency (<40%) and yield (<0.2 mmol h⁻¹ cm⁻²) are low, however, the result of this paper shows that Faradaic efficiency of acetamide is only 22%, and the yield is about 0.45 mmol h⁻¹ cm⁻². Moreover, due to the complexity of the reaction system, authors should conduct more in-depth research. Therefore I do not recommend its publication.

Response: We sincerely thank the reviewer for his/her critical comments on our manuscript. We must make a correction here that the 22% Faradaic efficiency refers **only to the anode half-reaction**. Since acetamide synthesis involves **both the cathode and anode processes**, the overall Faradaic efficiency for acetamide formation in the flow electrolyzer is **89%**. We have revised **Figure 4b** and relevant description to avoid this misunderstanding. We have also improved the detection of intermediates and the product stability test. Please find our point-by-point response below.

Question 1: Why are acetate products formed when methanol is used as a substrate in Figure S8 and Figure S9?

Response: Thanks for your pointing out these mistakes. There is a mistake in the captions of Figure S8 and S9 is wrong. We did ethanol oxidation rather methanol oxidation. We have revised them.

Question 2: The authors suggest that HNO is not only a reduction product in ethanol oxidation reaction, but also an important intermediate for ammonia production. Were the authors able to demonstrate the presence of HNO intermediate by spectroscopic or other characterization experiments?

Response: We are sorry that we cannot get clear evidence of HNO intermediate since it is highly reactive and a short-lived intermediate. We have tried our best to detect it through differential electrochemical mass spectrometry (DEMS), but no signal of HNO (m/z = 31) was found (**Figure R5**). Instead, NO has strong signal in DEMS and NO is an intermedia from HNO₂ to HNO. At the same current density (25 mA cm⁻²), ethanol addition to the NO₃RR system leads to a weakened NO signal on catalyst surface (m/z = 30, an intermediate from HNO₂ to HNO), supporting our proposed reaction mechanism (Figure S15).

Figure R5. DEMS validation of HNO intermediate (m/z =31) generation for NO₃RR with ethanol. No HNO signal was found.

Figure S15. DEMS validation of NO intermediate ($m/z = 30$) generation for NO_3RR with ethanol.

Question 3: *Since the test was carried out in the interval where there is no Faraday reaction, what is the cause of the current fluctuation in Figure S5?*

Response: Thank you for the comment. In Figure S5, the current fluctuations originate from unstable current under high scan rates. It can be seen that at low scan rate (20 mV s^{-1}), there is no current fluctuation. This test demonstrates that Rh single atoms modification leads to only a slight increase in the electrochemical active surface area (ECSA) of $\text{Ni}(\text{OH})_2$, which is not the primary factor contributing to the enhanced performance of the $\text{Rh}_1/\text{Ni}(\text{OH})_2$ catalyst. Current fluctuations do not affect this conclusion.

Question 4: *What is the Faraday side reaction of the cathode at low potential? Is there a reduction of acetaldehyde to ethanol?*

Response: Thank you for the comment. According to our previous study (10.1021/jacs.3c10516), at low potentials, nitrate reduction to nitrite is the dominant side reaction at the cathode. Under this condition, the total current remains low (less than 500 mA for a 16 cm^2 electrolyzer), leading to limited nitrite concentration and, consequently, insufficient ethanol oxidation to acetaldehyde. As noted in our response to Question 1 of Review 1, acetaldehyde is highly unstable in aqueous solution even without applied voltage, so there must be some acetaldehyde reduction to ethanol at cathode. Therefore, to enable effective ethanol oxidation to acetaldehyde via concentrated nitrite, and the subsequent formation of α -aminoethanol through reaction with concentrated ammonia, it is essential to operate the system in a flow electrolyzer at high current (total current $\geq 1 \text{ A}$).

Question 5: *The reaction system proposed by the authors requires the product produced by the cathode to enter the anode reaction. Does the rate of production of cathode products match the demand for anode? Is the insufficient supply of reactants the reason for the low Faradaic efficiency (22%) of anode target reaction?*

Response: Thank you for the comment. The α -aminoethanol yield rate from cathode has met the demand of anode, as evidenced by no α -aminoethanol signal in the ^1H NMR spectrum of the anode outlet electrolyte. The cathodic α -aminoethanol synthesis proceeds with a high Faradaic efficiency of 67%. In contrast, the 22% Faradaic efficiency at the anode reflects intrinsic limitations of the oxidation pathway:

As indicated by the reaction equation, our pathway involves a two-electron transfer at the anode and a six-electron transfer at the cathode. Therefore, the theoretical upper limit of the Faradaic efficiency for acetamide synthesis in our system is 33%.

The goal of this work is to establish a novel electrochemical pathway for amide synthesis by utilizing nitrate reduction. We have tried our best in enhancing the efficiency of this route, achieving a 67% Faradaic efficiency at the cathode. However, further improvement on the anode side remains challenging due to intrinsic limitations of the current pathway. In future studies, we aim to explore more strategies to make use of α -aminoethanol intermediates for boosting the anodic Faradaic efficiency.

Besides, we have revised **Figure 4b** to prevent potential misunderstandings. The updated figure reflects we achieved maximum Faradaic efficiency of **89%** for acetamide synthesis in the flow electrolyzer.

Question 6: *Could the authors quantify the α -aminoethanol produced by the cathode? Did the authors try to investigate anode reaction during α -aminoethanol as sole reactant?*

Response: Thank you for the comment. As explained in **Question 3 of Reviewer 1**, we cannot find reliable source of α -aminoethanol from major reagent suppliers (Sigma-Aldrich, Tokyo Chemical Industry, and BLDpharm). Also, the α -aminoethanol intermediate is inherently unstable and gradually decomposes in the electrolyte in a short time. Therefore, we cannot quantify α -aminoethanol and used it as sole reactant.

Question 7: *Will the ammonia produced by the cathode be oxidized when it enters the anode?*

Response: Thank you for the comment. In our system, there is a significant concentration difference between ammonia and ethanol, resulting in minimal ammonia oxidation.

At 2.4 V, the average current for acetamide synthesis is 1.73 A. If we assume all the electrons at cathode are used for ammonia synthesis (100% cathode Faradaic efficiency for ammonia), the ammonia concentration will be:

$$C_{\text{Ammonia}} = (i \times 60 \text{ s} \times FE_{\text{Ammonia}} \times 1000 \times 1000) \times (v \times 8\text{e}^- \times F \times 100\%)^{-1} = 89.7 \text{ mM}$$

(i is the cell current, v is the flow rate, F is Faraday constant)

Therefore, we evaluated the competition between ammonia oxidation and ethanol oxidation in H-cell with 15 mL electrolyte containing 1 M KOH, 0.05 M $(\text{NH}_4)_2\text{CO}_3$ and 1 M $\text{CH}_3\text{CH}_2\text{OH}$. $\text{Rh}_1/\text{Ni}(\text{OH})_2$ catalyst was used as working electrode. After 5 min reaction under 1.7 V vs. RHE (**Figure R6**), there is ~86 mM of ammonia remaining. If we assume that all ammonia is oxidized to nitrite, ammonia oxidation only accounts for ~20% of Faradaic efficiency.

In our tandem system under 2.4 V cell voltage, the concentration of generated ammonia in electrolyte is only ~30mM, so the ammonia oxidation should not be the key reaction.

Figure R6. Current density and ammonia concentration after reaction in H-cell. At each potential, the reaction lasts five minutes.

Question 8: Both 1 M KOH and 0.1 M KOH were used in the study. How does the concentration of KOH affect the reaction? Is acetamide stable in alkaline electrolytes?

Response: Thank you for the comment. We have prepared 50 mM acetamide solution at pH 13 (0.1 M KOH) and pH 14 (1 M KOH), and conducted NMR test within 10 minutes. As shown in **Figure S23**, acetamide is stable at pH 13 but decomposes into acetate and ammonia at pH 14. Therefore, we choose pH 13 for acetamide synthesis.

Figure S23. ¹H NMR validation of acetamide stability in aqueous solution with pH (a) 13 and (b) 14. If pH is

14, acetamide will gradually decompose into acetate and ammonia.

Revision:

1. We have added the sentence in manuscript: "Differential electrochemical mass spectrometry (DEMS) revealed that, at the same current density (25 mA cm^{-2}), ethanol addition to the NO_3RR system resulted in a smaller NO signal on catalyst surface ($m/z = 30$, an intermediate from HNO_2 to HNO), supporting our proposed reaction mechanism (Figure S15)" in Page 7.
2. We have revised the sentence into "As illustrated in **Figure 4a**, a high yield of acetamide (7.2 mmol h^{-1} , $0.45 \text{ mmol h}^{-1} \text{ cm}^{-2}$) was achieved at a cell voltage of 2.4 V, with 89% Faradaic efficiency (**Figure 4b**)."
3. We have revised **Figure 4b** and its caption in Page 10.
4. We have revised captions of Figure S8 and S9 in supporting information.

Reviewer #3:

*The authors reported that **a new synthesis avenue** to prepare acetamide by coupling nitrate electrochemical reduction and nitrite oxidation of ethanol. The new insights reported in this manuscript are **significant and attractive**, and may be further improved by focusing on or putting more emphasis on analysis of catalyst structure and product after continuous acetamide synthesis. Some revisions are required before the final publication:*

Response: We sincerely thank the reviewer for his/her positive comments on our manuscript. We have carefully revised the manuscript according to your suggestions on catalyst structure after reaction. We have also supplemented the product quantification and improved the discussion and experiment section parts. Please find our point-by-point response below.

Question 1: *To examine the structural integrity of the electrocatalyst, it would be benefited to conduct further structural analysis of $\text{Rh}_1/\text{Ni}(\text{OH})_2$ catalyst before and after continuous acetamide synthesis at 2.4 V cell voltage for 3000 minutes (Figure 4e). Has Rh single atoms aggregated? Has the $\text{Ni}(\text{OH})_2$ support corroded?*

Response: Thanks for your comment. Please refer to our response in **Question 8 of Reviewer 1**. We performed Rh K-edge EXAFS spectra for $\text{Rh}_1/\text{Ni}(\text{OH})_2$ catalyst after reaction to confirmed that Rh single atoms didn't aggregate and that the catalysts remained stable. We have also provided SEM images to prove that there are no noticeable morphological changes of $\text{Rh}_1/\text{Ni}(\text{OH})_2$ catalyst after the stability test.

Question 2: *What are the by-products in the continuous acetamide synthesis? How was the purified acetamide obtained?*

Response: Thanks for your comment. We have supplemented relevant description in the manuscript. As shown in **Figure 4b**, the by-products are ammonia from nitrate reduction and acetate from ethanol oxidation. The acetamide was purified through: 1) Vacuum drain the electrolyte at room temperature; 2) Add ethyl acetate to dissolve acetamide; 3) Vacuum drain the ethyl acetate solution at room temperature to obtain purified acetamide.

Question 3: *The selectivity of nitrate reduction to nitrite in the $\text{Ru}/\text{Cu}_2\text{O}$ cathode has not been quantified,*

which may affect the reaction efficiency.

Response: Thanks for your comment. We conducted NO₃RR at varying flow rates in pH 13 aqueous solution without ethanol addition, with a cell voltage of 2.4 V. While higher flow rates enhance the Faradaic efficiency for nitrite (Figure S27a), they also dilute nitrite and ammonia concentrations, leading to a notable decrease in acetamide yield (Figure S27b). Under this flow rate, the Faradaic efficiency of products (nitrite and ammonia) from NO₃RR without ethanol addition is shown in Figure S27c.

Figure S27. (a) Nitrite and ammonia and (b) acetamide synthesis performance at different flow rates (mL min⁻¹) for 16 cm² flow electrolyzer under 2.4 V cell voltage. 1.5 mL min⁻¹ is found to be the most suitable flow rate for acetamide synthesis. (c) Nitrite and ammonia Faradaic efficiency under different cell voltage for 16 cm² flow electrolyzer at 1.5 mL min⁻¹ flow rate.

Question 4: Figure 2e does not have any corresponding explanations in the manuscript.

Response: Thank you for the comment. We have added the explanation of Figure 2e:

“As shown in Figure 2d, this system achieves 9.4 A current at 2.4 V cell voltage, synthesizing ammonia at 90.7% Faradaic efficiency with 39.7 mmol h⁻¹ yield rate and acetate at 81.6% Faradaic efficiency 71.6 mmol h⁻¹ yield rate (Figure 2e and 2f), which is significantly higher than using Ni(OH)₂ as anode (Figure S9).”

Revision:

1. We have revised the sentence into “Various electrolyte flow rates were examined in our system. While higher flow rates enhance the Faradaic efficiency for nitrite (Figure S27a), they also dilute nitrite and ammonia concentrations, resulted in decreased acetamide yield (Figure S27b). Therefore, 1.5 mL min⁻¹ was identified as the optimal flow rate to fully utilize electro-generated nitrite and ammonia. Under this flow rate, the Faradaic efficiency of products (nitrite and ammonia) from NO₃RR without ethanol addition is shown in Figure S27c.” in Page 9.
2. We have revised the sentence into “We vacuum-drained the electrolyte at room temperature after the reaction, then extracted acetamide from the resulting solid using ethyl acetate. The extract was subsequently vacuum-dried again at room temperature to generate purified acetamide.” in Page 11.

Point-by-point responses to reviewers' comments for "Coupling Nitrate Electrochemical Reduction and Nitrite Oxidation of Ethanol for Acetamide Synthesis" (NCOMMS-25-17318A-Z)

Reviewer #1:

The author has addressed my questions well. However, I still have a minor concern, please refer to the following questions.

The experimental description presents some apparent contradictions regarding ethanol introduction timing and reaction initiation control. In Page 2, the authors stated that "Ethanol was added to the cathode during nitrate reduction, where it was oxidized to acetaldehyde by nitrite, an intermediate in nitrate reduction." However, Page 15 clarifies that the electrolyte (containing 0.1 M KOH, 1 M KNO₃ and 1 M CH₃CH₂OH) flows into the cathode first, then proceeds to the anode without further circulation. This suggests ethanol was present from the beginning rather than being added mid-reaction. Page 3 further notes: "We found that if the reaction starts from the anode, the production of acetate competes with acetaldehyde, resulting in a much poorer acetamide yield rate." This raises important questions about: (1) the precise mechanism ensuring cathode-first reaction initiation (as opposed to anode initiation), and (2) what reactions occur at the anode during the initial phase. The authors' repeated emphasis on the flow cell's special environment would be better supported by including a schematic diagram or a video to clarify these operational details.

Response: We sincerely thank the reviewer for his/her positive comments on our manuscript. We have revised the confusing sentences of introduction part in Page 2.

We also revised Figure 1 and provided Figure S1 to better demonstrate the difference between reactions initiated at the cathode and at the anode. If the ethanol oxidation initiates at the anode, there is a competing reaction of over-oxidation to acetate. Therefore, it is more advantageous for the coupled chemical oxidation (alcohol to aldehyde) and electrochemical reduction of aldehyde to acetamide at the cathode, as illustrated below.

Figure S1. Full cell reaction pathway for direct acetamide electrosynthesis (a) initiate at the cathode, (b) initiate at the anode.

Reviewer #2:

Authors have addressed reviewers' concerns largely. It could be accepted.

Response: We sincerely thank the reviewer for his/her positive comments on our manuscript.